∂ | **Open Peer Review** | Applied and Industrial Microbiology | Research Article

# A native bacterial consortium degrades estriol in domestic sewage and activated sludge via the 4,5-*seco* pathway and requires estriol to retain its biodegradation phenotype

Jaleela S. Hashem,[1] Wael Ismail,[2] Yin-Ru Chiang,[3] Vartul Sangal,[4] Dorra Hentati,[2] Nasser Abotalib,[2] Adnan A. Bekhit[1,5]

**ABSTRACT**   Estriol (E3) is one of the natural steroid estrogens commonly detected in wastewater. Although microbial biodegradation is a key process for removal of steroid estrogens during wastewater treatment, estriol biodegradation, and biotransformation mechanisms, as well as the involved bacterial consortia remain to be revealed. We enriched three E3-degrading bacterial consortia from raw sewage (inflow wastewater) and sludge samples. These consortia were able to utilize E3; however, individual strains isolated from the consortia could not grow on E3 as the sole carbon source. Instead, they transformed E3 to 16α-hydroxyestrone (16α-OH-E1) as a key product. The E3-transforming bacteria were affiliated with the genera *Hydrogenophaga*, *Microbacterium*, and *Gordonia*. The inflow (IF)-consortium utilized estrone (E1), estradiol (E2), in addition to E3 individually and as a mixture in minimal medium, raw sewage, and activated sludge microcosms, whereas the synthetic estrogen ethinylestradiol (EE2) was not degraded. Degradation of E3 was initiated by transformation to E1 via transient formation of 16α-OH-E1 and then proceeded via the 4,5-*seco* pathway. The community structure of the IF-consortium strongly shifted toward *Croceicoccus estronivorus,* which dominated the community after 10 days of incubation. The IF-consortium lost the E3 biodegradation phenotype upon growth on dimethylsulfoxide in the absence of estrogens. We conclude that complete degradation of E3 proceeds via the 4,5-*seco* pathway and requires concerted action of several community members of the IF-consortium at different time intervals and depending on the substrate and product concentrations. However, functional resilience of the consortium is a crucial factor that needs to be carefully addressed.

**IMPORTANCE**   Environmental pollution with endocrine-disrupting chemicals, including steroid hormones such as estriol, has become a global concern due to their hazardous impact both on aquatic life and human health. Elimination of estrogens from the environment occurs mainly via microbial biodegradation during wastewater treatment. However, efficient application of bioremediation requires thorough understanding of the structure and dynamics of estrogen-degrading microbial communities in wastewater and the underlying biodegradation and biotransformation mechanisms, which is currently lacking for estriol. In this study, we elucidated the estriol biodegradation pathway adopted by an estriol-degrading consortium obtained from raw domestic sewage. Furthermore, we unraveled how time and substrate type are determinants of the composition and the function of this consortium, which eliminated a mixture of natural estrogens in activated sludge and raw sewage. Hence, our findings constitute a step forward toward developing estrogen-degrading microbial consortia for more efficient removal of these pollutants during wastewater treatment.

Address correspondence to Wael Ismail, waelame@agu.edu.bh.

The authors declare no conflict of interest.

KEYWORDS estrogens, wastewater, biodegradation, activated sludge, consortia, estrone, estradiol

Steroid estrogens constitute a group of $C_{18}$ steroid hormones produced by humans and animals where they play key physiological roles in the regulation of the reproductive systems and development of secondary sex characteristics (1). However, chronic exposure to trace amounts of estrogens (sub-nanomolar levels) can disrupt the endocrine system and sexual development (2–5). These hazardous properties raised substantial concerns on environmental pollution with estrogens, which were classified by the WHO as group-1 carcinogens.

Steroid estrogens are represented by three naturally occurring hormones, including estrone (E1), estradiol (E2), and estriol (E3), as well as the synthetic estrogen ethinylestradiol (EE2) (6) (Fig. S1). They enter the environment via different routes, but mainly through discharge of urine and feces reaching wastewater treatment plants (WWTPs) (7–10). Estrogens are commonly present in wastewater in trace amounts, ranging from several ng/L to µg/L, yet they are associated with adverse ecological impact and influence marine organisms in the impacted area, causing acute and chronic toxicity to organisms, as well as loss of habitat and biodiversity (7, 11, 12).

Microbial biodegradation plays a key role in estrogen removal from WWTPs (12–14). Several estrogen-degrading bacteria have been isolated with focus on E1 and E2 (12, 15–17). On the contrary, knowledge on catabolic pathways and microbes involved in E3 biodegradation is very scarce. Most of the studies focused on pure cultures and degradation of single estrogen substrates in synthetic minimal medium, which is not necessarily environmentally relevant as wastewater contains mixtures of estrogens and other carbon sources (6, 18), Also, microbes exist as complex and heterogeneous communities or consortia in the environment. Some studies reported estrogen-degrading microbial consortia, but their functional stability was not studied (19–22), which is crucial for environmental applications in bioremediation.

Estriol, also called trihydroxy estrin and 16α-hydroxy estradiol, was discovered in the urine of a pregnant woman in 1930. Various concentrations of E3 and its conjugates were detected in influents and effluents of WWTPs in several countries (6, 18, 23–26). A few bacteria isolated from activated sludge or animal waste are capable of degrading E3 including *Acinetobacter* sp. (27), ammonia-oxidizing *Nitrosomonas europaea* (28), *Rhodococcus* sp. PI4 (29), *Comamonas testosteroni* (30), *Gordonia* sp. strain R9 (31), *Rhodococcus* sp. strain B50 (32), and *Pseudomonas putida* strain SJTE-1 (33). Except for transformation of E3 to 16α-OH-E1, no further degradation metabolites, enzymes, or genes were reported in those studies. Although E3 was detected as a metabolite of E2 degradation by *Gordonia* sp. strain R9 (31) and *Microbacterium resistens* MZT7 (34), knowledge of E3 degradation is still very limited.

A recent study by Liu et al. (35) proposed that E3 is first transformed to E1, followed by two potential biodegradation pathways. The first pathway involves degradation of E1 via the 4,5-*seco* pathway and the alternative pathway transforms E1 to 4-OH-E1, which undergoes B-ring cleavage, resembling the 9,10-*seco* pathway. However, the mechanism of transformation of E3 to E1 was not reported.

Here, we report the enrichment of three E3-degrading bacterial consortia from raw domestic sewage (inflow municipal wastewater), activated sludge, and thickened sludge samples collected from a major WWTP in Bahrain. We further characterized the inflow wastewater consortium (IF-consortium) in terms of substrate spectrum and bacterial community composition. We also tested the functionality of the consortium in raw sewage and activated sludge microcosms. Eventually, we elucidated the E3 degradation pathway and unraveled the role of E3 in maintaining the functional resilience of the IF-consortium.

## MATERIALS AND METHODS

### Collection of samples

The samples were collected from Tubli WWTP, the largest municipal WWTP in the east of Bahrain, next to the capital Manama (Fig. S2). It receives municipal wastewater from Manama, Muharaq, Isa Town, Hamad Town, Bani-Jamrah, and Budaiya, covering a population of 700,000 with a daily flow of 200,000–350,000 $m^3$ wastewater. Inflow wastewater (IF; raw sewage) samples were collected from the primary phase (screening phase). Activated sludge (AS) samples were collected from the secondary phase (aeration tank 2) and thickened sludge (TH) samples from the tertiary phase settling tanks. All samples (0.5 L) were collected in sterile 1 L glass bottles and transported to the laboratory within 30 min at ambient temperature (25–30°C). The samples were processed immediately to inoculate enrichment cultures.

### Enrichment of E3-degrading bacteria

A chemically defined medium (CDM) was used to enrich and isolate estrogen-degrading bacteria and to study estrogen biodegradation. The medium was prepared in deionized water and supplemented with 1 mM or 0.1 mM estrogen as a carbon source depending on the purpose of the experiments. For the enrichment culturing and initial screening of the isolates, 1 mM of E3 was added to provide sufficiently strong selective conditions. The inoculum source we used for enrichment culturing (sewage, sludge) naturally harbors heterogeneous and diverse communities of organisms as well as other organic constituents and suspended particles, which may interfere with the availability of E3 to the actual estrogen-degrading bacterial communities. Therefore, the addition of concentrations above the aqueous solubility limit (1 mM) in the enrichment cultures ensures sufficient supply of E3. For optical density measurements (growth profile experiments), 0.1 mM estrogens was used to avoid interference with the spectrophotometric measurements. The CDM consists of a basal buffer (ammonium chloride and phosphate buffer), which is complemented with vitamins, trace elements, iron, $CaCl_2$, and $MgSO_4$. The overall composition of the CDM per liter of deionized water was $KH_2PO_4$, 1.08 g; $K_2HPO_4$, 5.6 g; $NH_4Cl$, 0.54 g; $CaCl_2·2H_2O$, 0.044 g; $FeCl_2·4H_2O$, 1.5 mg; $MgSO_4·7H_2O$, 0.1 g; vitamins (cyanocobalamine 0.2 mg, pyridoxine-HCl 0.6 mg, thiamin-HCl 0.4 mg, nicotinic acid 0.4 mg, p-aminobenzoate 0.32 mg, biotin 0.04 mg, Ca-pantothenate 0.4 mg) and trace elements ($ZnCl_2·7H_2O$ 70 µg, $MnCl_2·4H_2O$ 100 µg, $CuCl_2$ 20 µg, $CoCl_2·6H_2O$ 200 µg, $Na_2MoO_4·2H_2O$ 40 µg, $NiCl_2·6H_2O$ 20 µg, $H_3BO_3$ 20 µg). Glucose, trace elements, and vitamins stock solutions were sterilized by filtration using 0.22 µm sterile filters (Stericups, Nalgene). The basal buffer was autoclaved, and the other components were added from sterile stock solutions. Liquid cultures were routinely prepared in 250 mL Erlenmeyer baffled flasks containing 100 mL of CDM and incubated at 30°C in an orbital shaker at 180 rpm.

To enrich E3-degrading bacteria, a 10 mL inoculum from the samples collected from the WWTP was added individually to CDM supplemented with 1 mM E3 (dissolved in DMSO) as a carbon source. After 7 days of incubation, 5 mL from initial enrichment cultures that showed growth (visually assessed by change in turbidity and or color) was transferred to new CDM cultures containing 1 mM E3 and incubated under the same conditions. This subculturing step was repeated three to five consecutive times. Samples from each enrichment were extracted and analyzed by GC-MS to verify the E3 biodegradation phenotype. Glycerol stocks of the initial enrichment and each subculture were stored at −80°C, and the three enrichments were designated as IF-consortium (enriched from inflow wastewater samples), AS-consortium (enriched from activated sludge samples), and TH-consortium (enriched from thickened sludge samples).

### Isolation of E3-degrading bacteria from the enriched consortia

To isolate E3-degrading pure cultures, samples (100 µL) from the third enrichment subculture were serially diluted in sterile normal saline, and aliquots of 100 µL from

the different dilutions were spread on LB agar plates and incubated for 2 days at 30°C. Exclusion of estriol from the LB medium was proposed to mitigate the selective pressure, thus enabling growth of both E3-degrading as well as E3-transforming bacteria. At the end of the incubation period, morphologically distinct colonies were selected and purified by three consecutive streaks on LB agar plates to obtain pure isolates.

## Growth of the isolated bacteria on E3

All isolates (total 39) were screened to reveal whether any of them can grow and utilize E3 as a carbon source. A single colony from each isolate was picked from the respective LB agar plates, inoculated into 20 mL LB broth medium and kept in the incubator for 1–3 days under the same incubation conditions. The biomass was harvested by centrifugation for 10 min at 4°C and 6,000 rpm, the cell pellet was resuspended in 10 mL complemented CDM (without E3), and the entire cell suspension was inoculated into CDM containing 1 mM E3. All cultures and uninoculated controls were incubated and checked visually at time intervals. Furthermore, culture samples were extracted and analyzed by GC-MS.

Following this initial screening, five isolates that showed E3 degradation potential were selected for further investigations. These isolates originated from different sources: strains TH-11, TH-16, and TH-30 were isolated from the TH-consortium, strain IF-20 from the IF-consortium, and strain AS-31 from the AS- consortium. They were cultured in CDM containing 0.1 mM E3 following the above-mentioned procedures. To test whether the selected five selected isolates can grow on E3 without adding the solvent DMSO, the same procedure was followed as described above to prepare inoculum, which was used to inoculate CDM containing 50 mg/L E3 as solid crystals. Growth was monitored by measuring $OD_{600}$ every 24 h, and culture samples were retrieved at time intervals to be analyzed by GC-MS.

## Identification of the isolates

Single colonies of the different isolates were picked from fresh LB-agar plates and suspended in 20 µL of sterile nuclease-free water in an Eppendorf tube. A 2 µL aliquot of the resulting cell suspension was then utilized in colony PCR for 16S rRNA gene amplification. The 16S rRNA gene was amplified using Taq PCR Master Mix (Qiagen, Germany) and 20 pmol of each universal primer: 27F (5′-AGAGTTTGATCCTGGCTCAG-3′) and 1492R (5′-GGTTACCTTGTTACGACTT-3′) in 100 µL reactions. The PCR cycles were initial denaturation at 94°C for 5 min; 30 cycles of denaturation at 94°C for 1 min, annealing at 54°C for 1 min, extension at 72°C for 1.5 min, and final elongation at 72°C for 10 min. Amplicons were analyzed by electrophoresis at 80 volts for 40 min on 1% agarose gels. The PCR products were purified using QIAquick PCR purification kit (Qiagen, Germany). The DNA was quantified using Nanodrop spectrophotometer (Thermo Scientific) and a Qubit 4 Fluorometer (Thermo Fischer Scientific).

## Preparation of precultures and the inoculum of the bacterial consortia

Precultures and inocula of the enriched bacterial consortia were routinely prepared from glycerol stocks frozen at −80°C. A frozen 2 mL vial was retrieved from −80°C and thawed on ice. Following gentle mixing, the content of the vial was centrifuged at 10,000 rpm for 5 min in a pre-cooled centrifuge at 4°C. The supernatant was discarded, and the resulting pellet was washed once with 1 mL of ice-cold 0.1 M phosphate buffer (pH 7), followed by centrifugation and decantation of the supernatant. The washed cell pellet was resuspended in 1 mL of the same buffer, and the cell suspension was divided into 0.5 mL portions to inoculate two flasks each containing 100 mL CDM supplemented with 0.1 mM E3. These cultures were then incubated at 30°C with shaking at 180 rpm for 5–6 days ($OD_{600}$ = 0.5–0.6). The biomass was harvested in 250 mL sterile centrifuge tubes at 10,000 rpm for 10 min at 4°C. The cell pellet was washed once with 25 mL of ice-cold 0.1 M phosphate buffer, transferred to a sterile 80 mL centrifuge tube, centrifuged again, and the pellet was resuspended in the same buffer to yield a homogeneous cell suspension.

This cell suspension was used as an inoculum, which was standardized to start with a biomass load of ~200 mg (dry cell weight)/L of culture medium.

## Growth profile of the enriched consortia and E3 biodegradation

The growth profile of each bacterial consortium on E3 dissolved in DMSO was studied in 100 mL CDM containing 0.1 mM E3. The precultures were prepared as described above, and from each preculture, 1 mL was used as an inoculum. Growth was monitored every 24 h by measuring the $OD_{600}$ until the cultures reached the stationary phase. Culture samples were retrieved, extracted, and analyzed by GC-MS. All cultures were run in biological triplicates. From the three enriched consortia, the IF-consortium was selected for further investigations including substrate spectrum, elucidation of the E3 biodegradation pathway, community structure analysis, functional stability, as well as functionality in sewage and activated sludge.

## Substrate spectrum of the IF-consortium

We selected the IF-consortium for further characterization because estrogen biodegradation by raw sewage-borne bacterial consortia has been rarely reported. We started by screening other steroid estrogens as growth substrates for the IF-consortium. The substrate spectrum of this consortium was studied in CDM containing other estrogens like E1, E2, or EE2 (0.1 mM each), individually and as a mixture with E3 (one mixture contained E1, E2, and E3, designated ME1, and another mixture contained E1, E2, E3, and EE2, designated ME2). In addition, growth of this consortium was tested on non-steroid carbon sources such as glucose (10 mM) and DMSO (1%, solvent of the steroid estrogens). The preculture and inoculum were prepared in CDM containing 0.1 mM E3 as described above. All cultures were grown in 100 mL CDM supplemented with the different substrates, inoculated with 1 mL of the inoculum cell suspension, and together with the abiotic controls were incubated under the same conditions. Growth was monitored by measuring the $OD_{600}$, and samples from the estrogen cultures were retrieved every 24 h and analyzed by GC-MS to monitor estrogen biodegradation.

## Elucidation the E3 degradation pathway and identification of the potential E3 degraders in the IF-consortium

A large-scale biodegradation experiment was conducted to elucidate the E3 degradation pathway and investigate temporal and substrate-associated compositional shifts in the structure of the IF-consortium to unravel potential key members involved in E3 degradation. The preculture and inoculum were prepared as described above, with a little modification where E3 concentration was 1 mM. The inoculum was suspended in 20 mL ice-cold phosphate buffer (pH 7), and 1 mL of this cell suspension was used to inoculate the different cultures.

The IF-consortium was cultured in 400 mL CDM (in 1 L Erlenmeyer baffled flasks) containing various carbon sources. Two cultures contained E1 or E3 (1 mM) and another culture contained 10 mM glucose and 1% DMSO. Glucose and E1 cultures were included to facilitate the identification of key E3-degrading members of the IF-consortium via comparative analysis of the temporal and substrate-dependent shifts in the community structure. All cultures were in biological triplicates and were incubated together with the abiotic controls under shaking (180 rpm) at 30°C for 14 days. The incubation period was extended to 14 days due to the high concentration of estrogens (1 mM) used in this experiment to ensure the accumulation of detectable amounts of the degradation intermediates (36). Culture samples (20 mL) were retrieved at day 0 (after 2 h incubation) and then every 48 h. From the collected 20 mL, 4 mL aliquots were transferred into 15 mL sterile tubes and kept at −80°C for DNA isolation and processing later, whereas the remaining 16 mL were kept at −20°C for extraction of the E3 degradation metabolites and chemical analysis.

## Identification of E3 degradation metabolites by UPLC-ESI-HRMS

The frozen samples (from both the E3 and E1 cultures) were thawed and mixed gently by inverting the tube 3–5 times. Then, 10 mL of each sample was mixed with 20 µL of EE2 stock solution in DMSO (as an internal standard) to a final concentration of 50 µM. The mixture was then transferred to a separating funnel and extracted with 10 mL of GC-MS grade ethylacetate. The organic phase was collected and evaporated in a rotary evaporator to reduce the volume to approximately 0.5 mL, followed by a complete dryness in a vacuum concentrator, and eventually stored at −20°C. All samples were analyzed by UPLC-ESI-HRMS on a UPLC system connected to an ESI mass spectrometer. Structural elucidation was performed as described (37–39).

## Analysis of compositional shifts in the IF-consortium

Seventy-two frozen samples were collected from the IF-consortium cultures grown on E1, E3, and glucose at different time points (every 48 h in triplicates, 8 time points) and stored at −80°C. The samples were thawed on ice and centrifuged for 10 min (at 4°C, 6,000 rpm). The supernatants were decanted, and the pellets were used to isolate total genomic DNA following the procedure described in the ab288102-Bacterial Genomic DNA Isolation kit (Abcam, UK).

The 16S rRNA amplicon sequencing was carried out by BIOTOOLS (https://www.tools-biotech.com/, Taiwan) using the circular consensus sequence (CCS) mode on a PacBio Sequel IIe instrument, followed by data analyses. PacBio SMRT Link software was used to select CCS reads with a minimum predicted accuracy of 0.9 and the minimum number of passes set to 3. Reads with ≥Q30 were analyzed by DADA2 version 1.20 for quality filtering, dereplication, application of the dataset-specific error model, inference of amplicon sequence variants (ASVs), and chimera removal (40, 41). The reads were trimmed and filtered with a maximum of two expected errors per read (maxEE = 2). DADA2 algorithm resolves ASVs with single-nucleotide resolution from the full-length 16S rRNA gene with very high accuracy. The number of reads after quality control varied between 9,123 and 20,045.

Taxonomic classifications for representative sequences were obtained from the NCBI 16S rRNA database using feature-classifier (42) and classify-consensus-vsearch (43) algorithms in QIIME2 v2022.11 (44). The sequence similarities among ASVs were analyzed using alignment tool MAFFT (45) within QIIME2 workflow. A phylogenetic tree was constructed for representative ASVs using the FastTree algorithm (46, 47). The QIIME package was used for calculating α and β diversities and R package ggplot2 was used to generate the plots. Bray-Curtis dissimilarity matrices were generated using microeco package in R. To visualize patterns in multidimensional data, dimensionality reduction methods NMDS (48) were used.

For statistical analysis, significance of all species among groups at various taxonomic levels was detected using differential abundance analysis with a zero-inflated Gaussian (ZIG) log-normal model as implemented in the "fitFeatureModel" function of the R Bioconductor metagenomeSeq package (49). The Welch's *t*-test was performed using the stat package in R. Statistically significant biomarkers were identified using the LEfSe analysis (50). For functional prediction analysis, 16S rRNA gene sequencing data were analyzed using FAPROTAX v1.2.6 (51). FAPROTAX is a manually constructed database based on the literature on cultured representatives. To find the important functional profiles across groups, R package ggplot2 and microeco were used to generate the plots and metastat statistics.

## Stability of the E3-degrading consortium

Although functional stability of microbial consortia is one of the crucial determinants of their applicability bioremediation, it is rarely addressed. Therefore, this experiment was performed to unveil whether the IF-consortium can retain its E3 biodegradation phenotype if transiently cultured in the absence of E3. Precultures of the IF-consortium

were prepared in CDM containing E3/DMSO, and after 5 days ($OD_{600}$, ~0.6), 1 mL was transferred into 100 mL CDM containing 0.1 or 1% autoclaved DMSO as the sole carbon source. Aliquots (1 mL) from those DMSO cultures were retrieved after 3 days and reinoculated in 100 mL CDM containing 0.1 mM E3/DMSO (duplicate cultures). All cultures and controls were incubated under the same conditions, and growth was monitored by measuring the $OD_{600}$ after 2 h and then at 24 h intervals for 8 days. Additionally, GC-MS analyses were conducted on samples collected from the E3-containng cultures on days 4, 5, and 12.

## Estrogen biodegradation in wastewater microcosms

As all the characterization experiments were done in synthetic minimal medium, we tested the functionality of the IF-consortium in real sewage and activated sludge. Therefore, biodegradability of estrogens, with and without bioaugmentation, was studied in microcosms containing either activated sludge (AS) or inflow (IF) wastewater samples and spiked with varying estrogen concentrations. Freshly collected samples from Tubli WWTP were transferred to the laboratory within 30 min at ambient temperature, where they were processed as soon as they reached the laboratory. Two different sets of microcosm experiments were performed, and in each set, three microcosms were prepared in 2 L sterile Erlenmeyer flasks each containing 500 mL of either AS or IF wastewater samples. The first set was designed to test whether the three natural steroid estrogens can be degraded by wastewater indigenous microbial communities, and the three microcosms of this set were (i) a microcosm spiked with E1, E2, and E3 (0.1 mM each); (ii) a microcosm spiked with E1, E2, and E3 (0.01 mM each); and (iii) a microcosm spiked with E1, E2, and E3 (0.1 mM each) after autoclaving the wastewater sample (abiotic control).

The second set of microcosms was similar to first set with the exception that the microcosms were autoclaved to eliminate the indigenous microflora and then bioaugmented with an inoculum from the IF-consortium (~200 mg dry cell weight/L). The inoculum of the IF-consortium was prepared in 400 mL CDM containing 0.1 mM E3 as described above. All microcosms were incubated at 30°C in an orbital shaker (180 rpm), and samples (20 mL) were retrieved after 2 h (day 0) and then every 48 h for GC-MS analysis.

Another microcosm experiment was conducted to study the biodegradation of EE2 by indigenous bacteria present in the inflow wastewater and activated sludge samples from the Tubli WWTP. This experiment was conducted in 1 L baffled Erlenmeyer flasks each containing 250 mL of freshly collected samples from the respective sources and spiked with 0.1 mM of EE2. All microcosms, including the controls (autoclaved), were incubated at 30°C in an orbital shaker at 180 rpm, and samples were collected after 2 h of incubation and subsequently at 48 h intervals, up to day 14. The samples were extracted and analyzed using GC-MS.

## Analysis of estrogen biodegradation by GC-MS

To investigate estrogen degradation by the isolated bacteria, mixed cultures (consortia), and wastewater microcosms, culture samples were analyzed by gas chromatography-mass spectrometry (GC-MS). Culture and microcosm samples (10 or 25 mL) were retrieved and extracted by equal volume of GC-grade ethylacetate. The organic phase was collected and concentrated to 1 mL in a rotary evaporator and analyzed by a Shimadzu gas chromatography system (GC 2010 plus) coupled to a Shimadzu MS-QP2020 mass detector, using a column Rxi-5SILms (30 m, 0.25 mm id, 0.4 µm df, Restek, USA). The oven temperature program was set to 1 min at 50°C with increments of 20°C/min up to 200°C where the oven was held at this temperature for 3 min, followed by 6°C/min up to 280°C, and finally, the temperature increased to 290°C with a rate of 30°C/min and held for 12 min. The GC was run in the split mode (split ratio 16.0) with a flow of 1 mL/min of helium as a carrier gas. The MS conditions were set to have the ion

source temperature at 250°C and the interface temperature at 280°C. The scan interval was 0.3 s, scan speed was 2,000, and the mass range was 50–600 *m/z*. The collected mass spectra were matched with the NIST spectral database (36).

## RESULTS

### Three E3-degrading bacterial consortia were enriched from raw sewage and sludge

Enrichment cultures grew on E3 as a carbon source in CDM inoculated with samples of raw sewage (inflow wastewater), activated sludge, or thickened sludge collected from a domestic WWTP. GC-MS analysis revealed E3 degradation within 4 days of incubation and those initial enrichments retained E3 biodegradation capacity even after repeated subculturing (Fig. S3 through S5). All three enrichments grew in E3-containing CDM (Fig. 1) when retrieved from frozen stocks, and GC-MS analysis confirmed E3 utilization within 4–5 days of incubation. Moreover, one degradation product was detected only in the thickened sludge cultures (Fig. S6). This product was designated product-1 and was detected in culture samples analyzed after 54 h of incubation. It appeared in the GC chromatogram before E3 at 34.5 min, indicating that its molecular mass is smaller than that of E3. Mass spectral database search revealed that product-1 has a molecular mass of 286 and the base peak appeared at *m/z* 213. This indicates that product-1 is two mass units smaller than E3.

Based on these criteria, it was assumed that product-1 is an oxidation/dehydrogenation product of E3. Indeed, mass spectral library search revealed the highest matches are oxidation products of E3 where two hydrogens are lost from the OH- group on either C16 or C17 (Fig. S7), thus predicting product-1 to be 16-keto-E2 or 16α-OH-E1, respectively.

### Isolation and identification of E3-transforming bacteria from the enriched consortia

To determine whether E3 biodegradation is a coordinated phenotype involving multiple members of the microbial consortium, we isolated individual strains and tested their E3

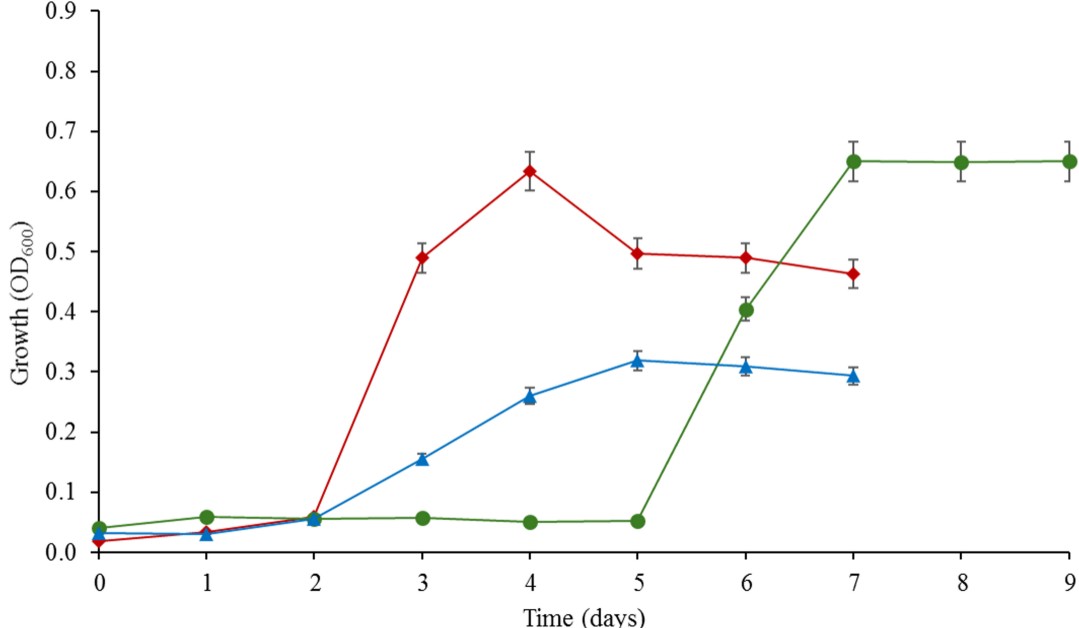

**FIG 1** Growth profiles of the E3-degrading bacterial consortia enriched from inflow wastewater (red line), activated sludge (blue line), and thickened sludge (green line). The cultures were grown in 100 mL CDM containing 0.1 mM E3 at 30°C with shaking at 180 rpm. The results are presented as averages of measurements from three biological replicates, and the error bars represent standard error. Estriol was added from a stock solution in DMSO.

degradation capacity. Out of the 49 isolates obtained from the three consortia, only five isolates transformed E3 into product-1 as a main E3 transformation product (Fig. S6, S8 through S12). Three of the isolates (TH-11, TH-16, TH-30) belonged to the TH-consortium, one isolate (IF-20) belonged to the IF-consortium, and the last one (AS-31) was isolated from the AS-consortium. However, none of the isolates could grow in CDM containing E3/DMSO even after 14 days of incubation, confirming that these isolates cannot utilize E3 or even the solvent as a carbon source. Exceptionally, isolate TH-16 exhibited a little increase of the $OD_{600}$ from 0.1 to 0.34 after 7 days, followed by a stationary phase (Fig. S13A). Assuming that DMSO could have an inhibitory effect on the growth of the isolates, they were cultured in CDM supplemented with E3 as solid crystals (without DMSO). The growth profiles were similar to those observed in cultures containing E3/DMSO (Fig. S13B), and all cultures transformed E3 into product-1.

Genomic DNA was extracted from the five E3-transforming isolates and used as a template to amplify the 16S rRNA gene (Fig. S14). Phylogenetic analysis confirmed the identity of the isolates (Fig. 2). The IF-20 isolate was most closely related to *Microbacterium* spp. with the highest identity (99.16%) to *Microbacterium schleiferi* DSM 20489, whereas the AS-31 isolate was 98.48% identical to *Hydrogenophaga intermedia* DSMZ 5680. On the other hand, all three isolates obtained from the TH-consortium were affiliated with the genus *Gordonia* and specifically to *G. cholesterolivorans* and *G. sehwensis*, with the isolate TH-16 exhibiting the lowest sequence identities of 97.29%–98.53%, respectively,

## Substrate spectrum of the IF-consortium

In addition to E3, the IF-consortium grew in CDM containing glucose, E1, E2, a mixture (ME1) of E1, E2, and E3, a mixture (ME2) of E1, E2, E3, EE2, or DMSO as a carbon source (Fig. 3). The ability of the inflow consortium to grow on DMSO as a sole carbon source raised the question whether the tested estrogens were utilized. GC-MS analysis confirmed the utilization of E1, E2, and E3 when added as a mixture (Fig. S15) or individually, and E2 was transformed to E1 to be further metabolized via the 4,5-*seco* pathway (37) (Fig. S16 and S17). On the contrary, EE2 was not consumed even after 60 days of incubation (Fig. S18), indicating that growth observed in the EE2 cultures was due to DMSO utilization. In contrast, when the four estrogens were added as a mixture, only E2 and E3 were consumed, and product-1 was detected. However, E1 and EE2 persisted up to 60 days of incubation (Fig. S19).

## The E3 biodegradation pathway

In the E3 cultures of the IF-consortium, four estrogen degradation metabolites were detected, namely, E1, 4-OH-E1, pyridinestrone acid (PEA), and HIP (3aα-H-4α(3′-propanoate)-7aβ-methylhexahydro-1,5-indanedione) (Fig. 4A). The concentration of E3 decreased with time, and by day 14, almost 80% of the added E3 was consumed (Fig. 4B). This drop in E3 concentration was accompanied by an increase of E1 concentration, which was detected after 6 days of incubation at a concentration of 0.01 mM, increasing to 0.61 mM at day 14. All the other intermediates were detected in much lower concentrations only starting from day 10 (Fig. 4B). Like the E1 profile, the concentration of 4-OH-E1and PEA increased from day 10 to day 14, with PEA being detected in lower concentrations compared to 4-OH-E1. In contrast, HIP could be detected only after 12 days of incubation, and its concentration did not change by day 14. The HIP concentration was lowest among the detected degradation metabolites.

In the E1 cultures, E1 concentration decreased with time, and by day 14, almost 50% was consumed (Fig. 4C). This drop in E1 concentration was accompanied by the formation of 4-OH-E1, which was detected after 4 days of incubation at 0.01 mM, increasing with time to reach 0.11 mM after 14 days. On day 6, PEA was detected, and its concentration increased with time to reach 0.05 mM after 14 days. The next intermediate, HIP, was detected in the E1 cultures after 8 days, and its maximum concentration was

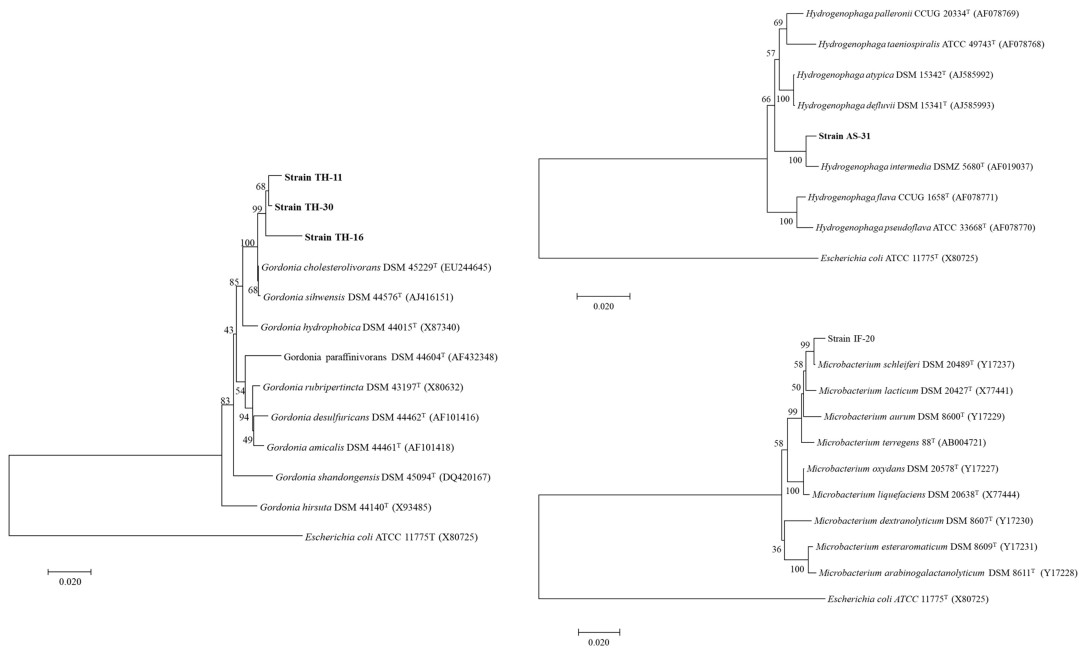

**FIG 2** Phylogenetic trees based on 1,431 bp of the 16S rRNA gene sequences obtained from the E3-transforming isolates. Reference type strains are included, and GenBank sequence accession numbers are given between parentheses. *Escherichia coli* was used as an out group. Bootstrap values, expressed as percentage of 1,000 replications, are given at the nodes. Bar: two substitutions per 100 nucleotides.

0.03 mM attained after 14 days. In both the E3 and E1 cultures, HIP appeared the last and its concentration was mostly lower than the other detected metabolites. On the contrary, the degradation metabolites appeared earlier in the E1 cultures, and their concentrations were generally higher compared to the E3 cultures. In the abiotic (negative) controls, no remarkable changes in the E1/E3 concentration were detected (Table S1).

These findings indicate that E3, similar to E1, is degraded by the IF-consortium via the 4,5-*seco* pathway (Fig. 5). Moreover, it can be postulated that the initial step of E3 degradation proceeds by transforming E3 to E1, probably via dehydroxylation/dehydration at C16 to produce E2, which is then oxidized to E1. Alternatively, E3 could be first oxidized to 16α-OH-E1 followed by removal of the OH group at C16 to produce E1. In either case, a dehydroxylase/dehydratase and dehydrogenase are likely to be involved.

## Analysis of the IF-consortium community structure during E3 biodegradation

To identify, at least tentatively, key E3-degrading bacteria in the IF-consortium, samples from E3 and E1 cultures, which were used to identify the degradation metabolites, were collected and analyzed at time intervals to study how the composition of the consortium changes with time and if the type of the estrogen substrate can affect the community structure. Furthermore, glucose cultures of the IF-consortium were run in parallel and analyzed to reveal the community involved in the utilization of non-estrogen substrate, which may facilitate more specific fingerprinting of the estrogen degradation-related community.

The microbial communities were analyzed every 48 h, starting from day 2, day 4, or day 10, for the glucose, E3, and E1 cultures, respectively. The microbial communities in the three cultures varied in their structure and prevalence of key taxa, depending on the carbon source. However, most species belonged to the phylum *Pseudomonadota* (formerly *Proteobacteria*) in all three cultures with species of phylum *Actinomycetota* (formerly *Actinobacteria*) being only prevalent in the E1 and E3 cultures (Fig. S20).

*Achromobacter insuavis*, *Achromobacter pulmonis*, *Croceicoccus estronivorus,* and *Hyphomicrobium denitrificans* were common to the E1 and E3 cultures (Fig. 6A). *A.*

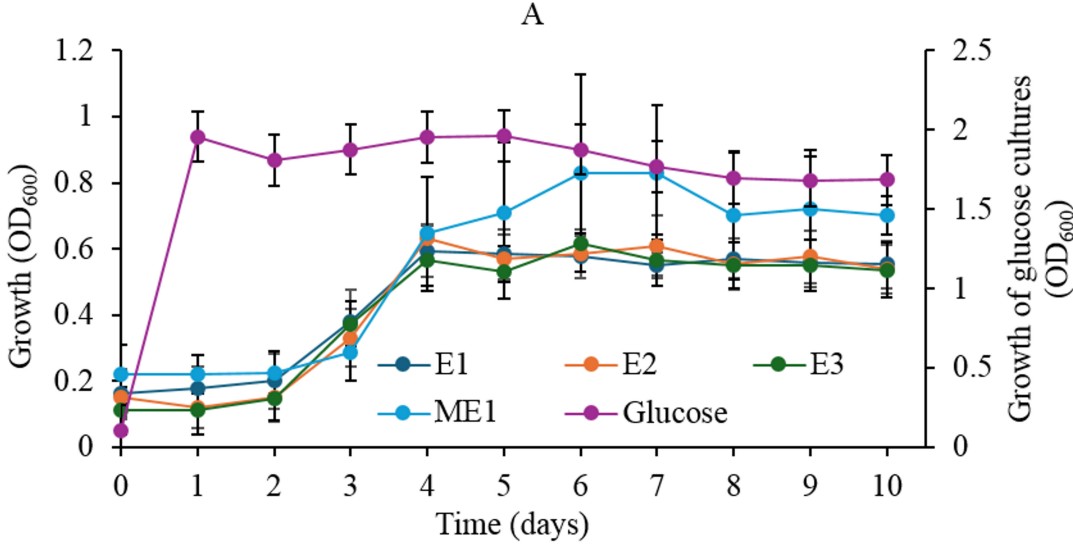

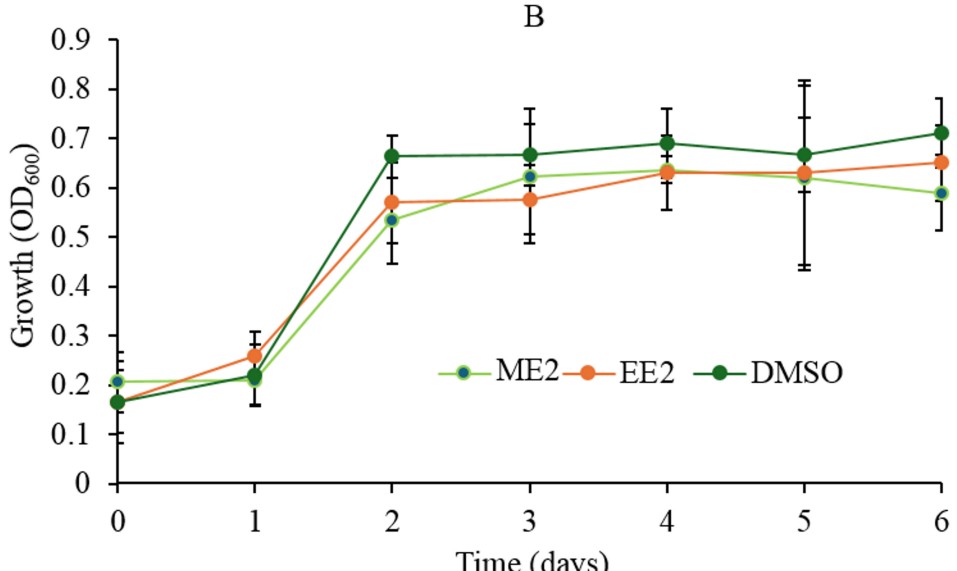

**FIG 3** (A) Growth profiles showing substrate spectrum of the IF-consortium in 100 mL CDM containing glucose (10 mM) or natural estrogens (0.1 mM each) individually and as a mixture (ME1: E1, E2, E3). (B) Growth profiles of the IF-consortium on DMSO (1%), EE2 (0.1 mM), or a mixture of estrogens including the synthetic estrogen (ME2: E1, E2, E3, and EE2). The results are presented as averages of measurements from three biological replicates, and the error bars represent standard error. Estrogens were added from stock solutions in DMSO.

*insuavis*, *A. pulmonis,* and *H. denitrificans* were the most prevalent taxa in the E3 cultures up to day 8 but were taken over by *C. estronivorus* from day 10 (Fig. 6B). *C. estronivorus* was most prevalent in E1 cultures; however, it is unclear if it replaced *A. insuavis*, *A. pulmonis,* and *H. denitrificans* starting from day 10, like the E3 cultures. Unfortunately, enough DNA required for sequencing was only obtained from the E1 cultures starting from day 10, and the distribution of other taxa between days 0 and 8 could not be determined. The dominant taxa detected in the E1 and E3 cultures were mostly absent in the glucose (G) cultures, except for *A. insuavis* that was observed across all glucose cultures, and *H. denitrificans* was only observed in a single glucose culture (Fig. 6B). Three *Klebsiella* species (*K. africana, K. quasipneumoniae,* and *K. pneumoniae*), *Citrobacter sedlakii,* and *Enterobacter cloacae* were prevalent among the glucose cultures.

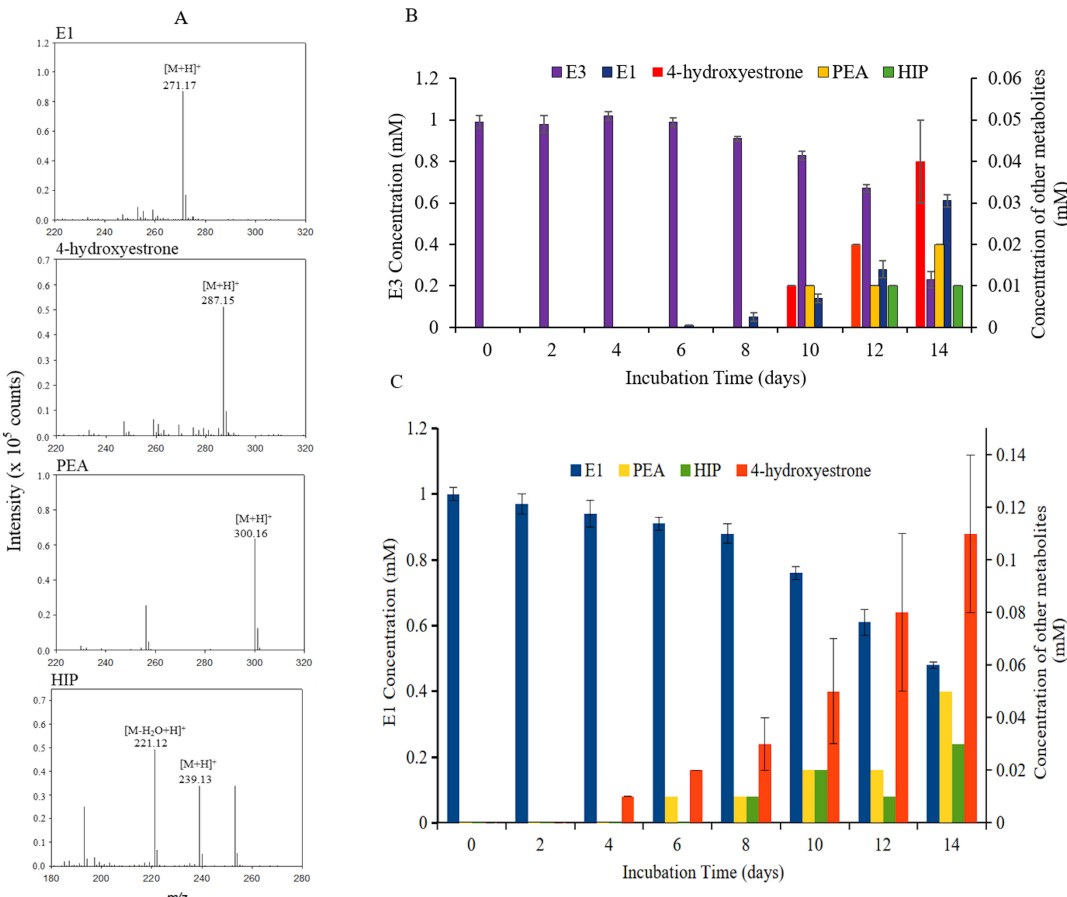

**FIG 4** UPLC-ESI-HRMS analysis showing mass spectra of metabolites detected in E3 cultures of the IF-consortium (A). Concentrations of the substrates and metabolites detected in the E3 (B) and E1 (C) cultures of the IF-consortium. The predicted elemental composition of individual metabolite ions was calculated using MassLynx Mass Spectrometry software (Waters).

In total, 43 (25.1%) amplicon sequence variants (ASVs) were shared between the three groups, 26.3% ASVs were specific to the glucose (G) cultures, and 23.4% to E3 cultures. Interestingly, only 1.8% ASVs were unique to the E1 cultures (Fig. 7A). The ASVs from the phylum *Pseudomonadota* were prevalent across all three groups, but those from *Actinomycetota* were more common in E1 and E3 cultures (Fig. 7B). It is worth noting that DADA2 algorithm defined ASVs based on single-nucleotide difference in the full-length 16S rRNA gene, resulting in multiple ASVs within each species. Therefore, we focused on the species that were unique to each culture. While *Leucobacter massiliensis* was unique to the E1 cultures, *Klebsiella pneumoniae* was exclusively present in the glucose cultures, and in only one E3 culture (E3.4b) (Fig. 7C). No species were found to be specific to the E3 cultures. Microbial heterogeneity within each group (culture) significantly differed from each other (Fig. 8A and B). The E3 culture samples contained more diverse microbial taxa in comparison to the E1 and glucose cultures, that latter was highly homogenous and dominated by *Klebsiella pneumoniae*.

The rarefaction curves showed that ASVs have plateaued at around 2,500 reads (Fig. S21), suggesting that most of the microbial diversity present in the samples was captured in this study. These results are highly robust showing that the microbial communities' structures varied significantly based on the substrate used in the growth medium; however, multiple microbial taxa were common between the E1 and E3 cultures (Fig. 8C).

LEfSe (Linear discriminant analysis effect size) was performed to identify distribution of significantly different taxa between the three cultures. This approach

**FIG 5** Proposed pathway for E3 biodegradation by the IF-consortium.

is extensively used in identifying biomarkers from metagenomic data. As previously observed, the taxa belonging to the phylum *Pseudomonadota* (particularly γ-*Proteobacteria*) were significantly more abundant in the glucose cultures including *K. pneumoniae* subsp. *rhinoscleromatis* and *similipneumoniae*, *K. Africana*, *Citrobacter sedlakii,* and *Enterobacter cloacae* subsp. *dissolvens*. Members of the order *Sphingomonadales*, families *Erythrobacteraceae* and *Xanthomonadaceae,* and the genera *Croceicoccus* and *Thermomonas* were prevalent in the E1 cultures, whereas the E3 cultures harbored a more diverse community including taxa belonging to the orders *Hyphomicrobiales*, *Micrococcales,* and *Xanthomonadales*, families *Alcaligenaceae*, *Burkholderiaceae*, *Rhodanobacteraceae*, *Phyllobacteriaceae*, *Hyphomicrobiaceae,* and *Microbacteriaceae,* and genera *Achromobacter*, *Aminobacter*, *Cupriavidus*, *Dyella*, *Hyphomicrobium,* and *Microbacterium*. However, taxonomic prevalence in the E3 culture switched after day 10, and prevalent taxa were similar to those observed in the E1 cultures, especially *C. estronivorus* (Fig. S22).

FAPROTAX software was used in converting taxonomic microbial community profiles into putative functional profiles. The taxa identified in all three cultures are aerobic chemoheterotrophs (Fig. S23 and S24). However, the dominant taxa in the glucose cultures have more capacity to reduce nitrate and ferment sugars. These taxa also have pathogenic capacities, including potential to cause pneumonia and septicemia in humans. In contrast, bacteria dominating the E1 and E3 cultures are capable of ureolysis, nitrate respiration, and oxidation of various sulfate sources.

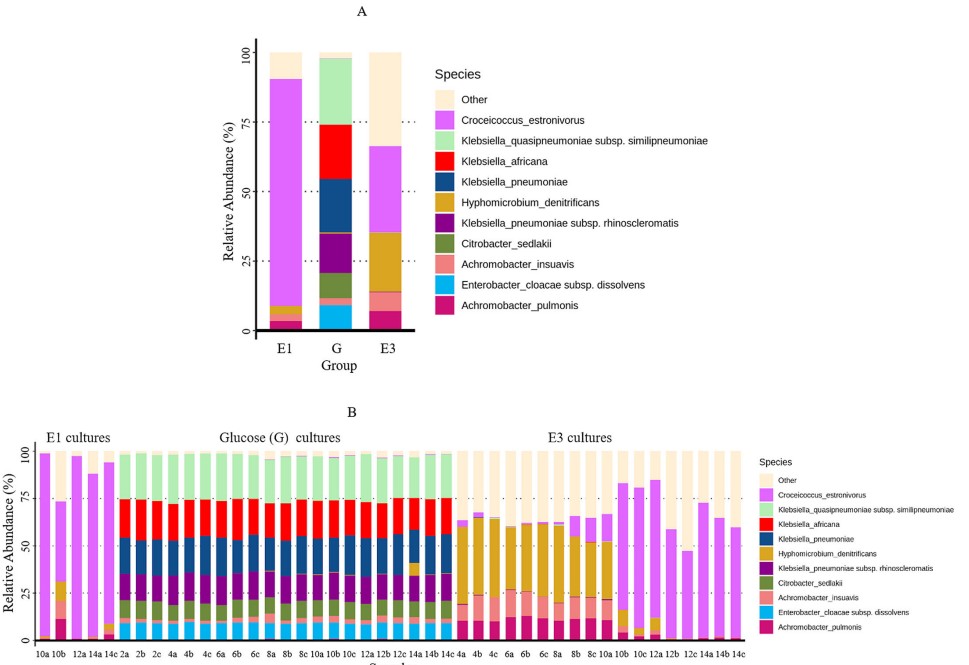

**FIG 6** Average relative abundance of top 10 species identified among the IF-consortium cultures across all time points (A) and in individual samples analyzed at different time intervals (B). Estrone (E1), glucose (G), and estriol (E3) were used as carbon sources. Numbers of the sample labels indicate the incubation period in days, and the letters designate the biological replicate.

## Estrogen biodegradation in wastewater microcosms

The GC-MS analysis revealed that the three estrogens E1, E2, and E3 were degraded in both activated sludge and inflow wastewater microcosms at 1 mM or 0.1 mM final concentration (Fig. S25 and S26). After 2 days of incubation, no estrogens could be detected in either microcosm. No degradation products could be detected even after 2 weeks of incubation, except product-1, which was detected only in the activated sludge microcosms after 2 h of incubation and subsequently disappeared. To unravel whether the IF-consortium can degrade estrogens in wastewater samples, it was prepared as an inoculum and added to autoclaved inflow wastewater and activated sludge microcosms containing a mixture of the natural estrogens E1, E2, and E3. These estrogens were degraded and consumed within 2 days of incubation, indicating the functionality of the inflow consortium in real wastewater samples. As observed in the previous experiment (without augmentation), no degradation products could be detected (Fig. S27 through S30). On the contrary, EE2 did not show any detectable signs of degradation or transformation after 14 days of incubation in activated sludge or inflow wastewater microcosms.

## Stability of the E3-degrading IF-consortium

To check the stability of IF-consortium, an inoculum from E3 (dissolved in DMSO) cultures was grown in CDM with two different DMSO concentrations (1% and 0.1%). The IF-consortium grew in this medium, and the growth profiles of both cultures were very similar, indicating the ability of the IF-consortium to utilize DMSO as a carbon and energy source (Fig. S31A). When aliquots from those DMSO cultures were transferred to fresh CDM containing E3/DMSO, the cultures grew, and their growth profiles were similar to those of the DMSO cultures (Fig. S31B). However, E3 was not degraded in these cultures even after 12 days of incubation (Fig. S32), suggesting that the IF-consortium lost the E3 degradation phenotype when grown in the absence of E3.

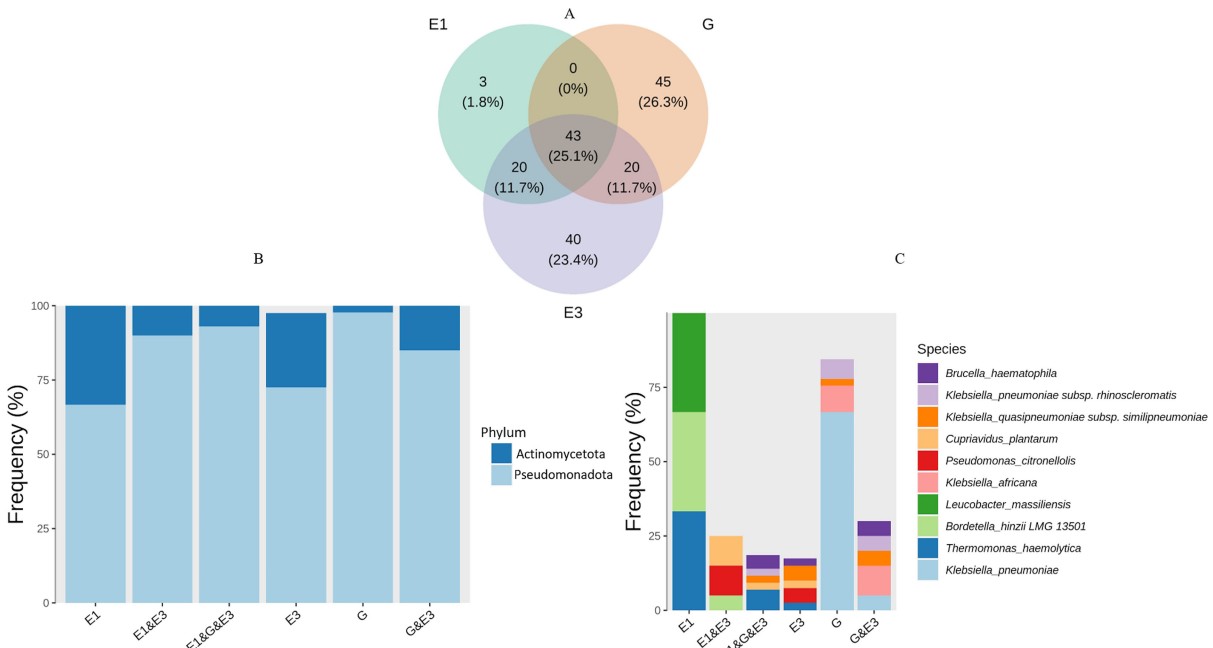

**FIG 7** Venn diagram showing the number of ASVs shared between the three cultures (A) and frequency of shared ASVs at phylum (B) and species level (C). E1 (estrone), G (glucose), and E3 (estriol) were used as carbon sources.

## DISCUSSION

Our findings addressed some of the knowledge gaps related to E3 biodegradation, which shall pave the way for further development studies. The elucidation of a proposed biodegradation pathway and the structural/functional analyses of the IF-consortium could fuel efforts aiming at the development or construction of efficient wastewater bioremediation processes tailored for estrogen elimination. We also drew attention to the importance of investigating the functional stability of the constructed or native consortia, a criterion that is rarely addressed in the relevant literature.

The ability of the three enriched bacterial consortia to retain E3 degradation capability after repeated subculturing of the initial enrichments indicated utilization of E3 as a carbon and energy source. Further evidence for mineralization of E3 by the enriched consortia could be envisaged from the ability of the three consortia to grow in cultures containing E3, which was consumed over time as shown by GC-MS analysis. Formation of product-1 in the cultures of the TH-consortium and cultures of isolated bacteria suggested a key role in E3 metabolism. According to mass spectral search, it appears that product-1 is likely 16α-OH-E1, potentially resulting from E3 by oxidation or dehydrogenation of the C17 hydroxyl group, a reaction that is catalyzed by 17β-hydroxysteroid dehydrogenase (6, 29).

These findings are in line with those reported by Ke et al. (27) where an *Acinetobacter* sp. degraded E3 as a sole carbon source and transiently produced 16α-OH-E1, which was also reported as a metabolite of E2 degradation by sewage bacteria (52) and of E1 degradation by intestinal bacteria (53). Moreover, 16α-OH-E1 can be reduced to E3 under anaerobic conditions by *Staphylococcus aureus* (53). At this stage, it was postulated that 16α-OH-E1 is a key intermediate in the E3 degradation pathway. The majority of E3-degrading bacteria reported to date were pure cultures and were isolated mainly from activated sludge, farmland soil, compost, and animal waste (15, 31, 54). To the best of our knowledge, this is the first study reporting E3-degrading consortia. Consistent with our findings, some studies reported E2 degradation by *Microbacterium* sp. (34, 55, 56), and *Gordonia* sp. degraded the four estrogens as a sole carbon source (31), cholesterol (57), and bile acids (58). *Hydrogenophaga* spp. were also implicated in the

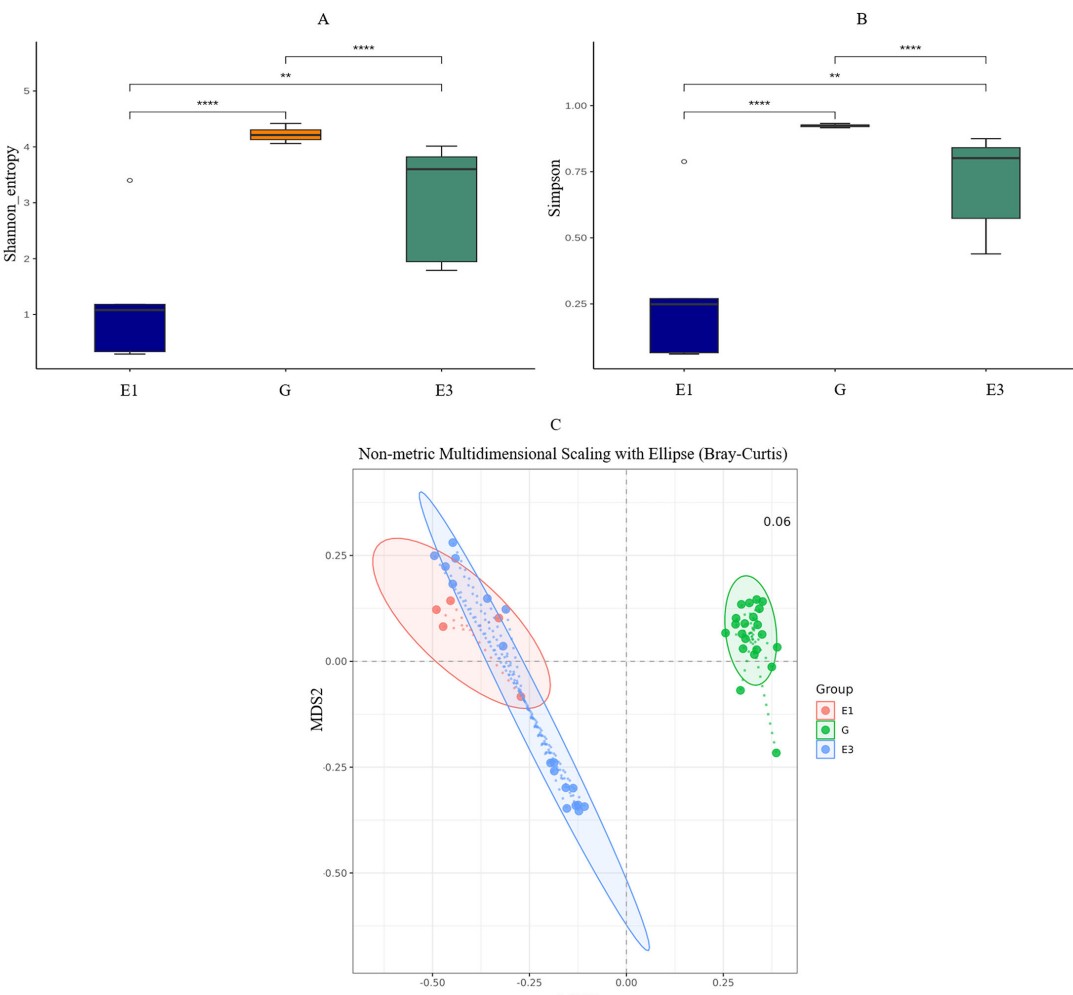

**FIG 8** The α-diversity between the three groups using Shannon index (A) and Simpson index method (B), and the β-diversity showing differences in microbial community composition between the three groups (C). E1 (estrone), G (glucose), and E3 (estriol) were used as carbon sources.

degradation of E1 (59), bisphenols (60), sulfamethoxazole (61), and phthalic acid (62). Moreover, *Hydrogenophaga* spp. were reported among the dominant members of a bacterial community in a membrane bioreactor treating synthetic hospital wastewater (63).

Growth profiles and GC-MS analysis confirmed the ability of the IF-consortium to grow on and utilize the natural estrogens E1 and E2 either individually or in a mixture with E3. Formation of product-1 in cultures containing a mixture of the four estrogens was concomitant with E3 degradation, further confirming that product-1 is an intermediate of E3 biodegradation. Furthermore, utilization of E2 in this culture explains why E1 increased with time, which is in line with the reported metabolic transformation of E2 via E1 (38).

Elimination of estrogen mixtures at two different concentrations confirmed the functionality of the IF-consortium not only in raw domestic sewage from which the consortium was initially obtained but also in activated sludge. While biodegradation of estrogens in activated sludge microcosms and by mixed cultures enriched from activated sludge has been studied (19–21, 64), similar studies for raw domestic sewage (inflow wastewater) and particularly for E3 biodegradation are largely lacking. However, in most of those studies, the enriched consortia or isolated bacteria were not tested to reveal their functionality when re-inoculated into activated sludge samples.

Analysis of the E3 and E1 degradation metabolites by UPLC-HRMS provided two key clues. The first confirmed that E1 and E3 were mineralized as carbon sources in the IF-consortium cultures despite the presence of DMSO. The second clearly proves that E3 and E1 are metabolized via the 4,5-*seco* pathway. Although E1 metabolism via the 4,5-*seco* pathway was reported by some bacteria (36, 38, 39), this was not the case for E3 until very recently (35). The detection of E1 as a key metabolite in the E3-grown cultures of the IF-consortium suggested that transformation of E3 to E1 is one of the initial reactions in the E3 catabolic pathway. We can envisage several possible scenarios on how E3 can be transformed to E1. Based on our findings, E3 is first oxidized by a putative hydroxysteroid dehydrogenase at either C17 or C16 to produce 16α-OH-E1 or 16-keto-E2, respectively (product-1) (6, 27, 29).

As we did not detect any E2 derivatives, we postulate that product-1 is 16α-OH-E1. Further metabolism of product-1 via D-ring cleavage (6) can be excluded as we did not detect D-ring cleavage products. In contrast, the detection of E1 and 4-OH-E1 strongly suggests that product-1(16α-OH-E1) is transformed to E1 via removal of the C16 hydroxyl group by a putative dehydroxylase/dehydratase, followed by further metabolism via the 4,5-*seco* pathway (35).

Although the involvement of a dehydroxylation reaction in aerobic or anaerobic microbial estrogen metabolism has not been reported (to our knowledge), biological dehydroxylation is known to occur in bile acids metabolism by gut-associated bacteria and via the Hylemon-Björkhem 7α-dehydroxylase pathway of bile acid degradation (65, 66). Moreover, biological dehydroxylation is a key step in the bacterial anaerobic degradation of 4-hydroxybenzoyl-CoA and other aromatic compounds which are metabolized via 4-hydroxybenzoyl-CoA such as phenol, 4-hydroxybenzoate, *p*-cresol, and 4-hydroxyphenylacetate (67), in a Birch-reduction-like mechanism.

The remarkable temporal differences between the glucose cultures and the estrogen-containing cultures indicated uniqueness of the estrogen-degrading community whose members were strongly enriched in the presence of estrogens (21, 64). Considering that the inoculum used for this experiment was prepared in an E3-containing culture, it can be postulated that the original composition of the IF-consortium is similar to the composition of the consortium revealed at day 4 in the E3 cultures because data for day 0 and day 2 are missing. Hence, the IF-consortium originally consisted mainly of members of the genera *Hyphomicrobium*, *Achromobacter*, *Microbacterium*, and *Cupriavidus* as the most dominant members. This community structure may constitute the key E3-degrading community up to day 10, where the community became strongly dominated by *Croceicoccus estronivorus*, an α-proteobacterial member of the *Erythrobacteraceae* (68), which could potentially be a key E3 degrader during the late stages of the culture. Nonetheless, these assumptions await validation.

The detection of *Microbacterium schleiferi* as one of the dominant members in the E3 culture is consistent with the isolation of *Microbacterium schleiferi* DSM 20489 (IF-20) from the IF-consortium grown on E3 and suggests a role in E3 metabolism. Among the dominant bacteria identified in the estrogen-grown cultures, members of the genera *Achromobacter*, *Microbacterium*, and *Aminobacter* were reported in estrogen biodegradation studies (15). Also, *Hyphomicrobium* spp. were detected among the dominant members of estrogen-degrading consortia enriched from activated sludge (69). However, *Croceicoccus estronivorus* (formerly, *Altererythrobacter estronivorus*) has been very rarely reported in estrogen biodegradation studies (70). *Croceicoccus* spp. were studied for the biodegradation of bisphenol-A (an endocrine disruptor) (54) and hydrocarbons (71). In mudflat sediment microcosms amended with bisphenol A, the relative proportion of *Croceicoccus* substantially increased with time, suggesting a role in bisphenol A biodegradation (72). Putative estrogen degradation genes were identified in *Croceicoccus* (*Altererythrobacter*) *estronivorus* (37, 73).

One of the crucial issues challenging the application of mixed cultures or microbial consortia in bioremediation is the functional resilience/resistance, i.e., stability of the consortium. Nonetheless, this key criterion was not properly studied for

estrogen-degrading bacterial consortia reported in the literature. Persistence of E3 up to day 12 indicated that the IF-consortium lost its E3 biodegradation phenotype. Loss of function suggests that the IF-consortium is not sufficiently resilient. It appears that growth on DMSO alone leads to extinction or elimination of the key E3 degraders from the consortium, which explains why the IF-consortium could not degrade E3 when transferred from the DMSO to the E3/DMSO cultures. It further indicates that the growth observed in the E3/DMSO cultures was due to DMSO utilization. DMSO biodegradation by bacteria was reported (74). Furthermore, it can be postulated that the key E3-degrading members are not able to metabolize DMSO as a sole carbon source, which explains why they need the presence of E3 to be retained in the consortium, i.e., E3 provides selective advantage to the consortium to maintain the E3 biodegradation phenotype even in the presence of other carbon sources.

Although the adopted experimental design and methods addressed the research questions, some underlying limitations need to be rectified in future studies. For instance, the presence of the estrogen solvent DMSO challenged concluding that estrogens were utilized as the sole carbon source. In addition, it may exert some toxic effect on the isolated microbial communities. The adopted estrogen concentrations were several orders of magnitude higher than the reported environmental concentrations in domestic sewage. This could select microbial communities, which are not the actual estrogen degraders in natural and engineered polluted environments. It can also lead to molecular and biochemical conclusions, which may not be necessarily applicable to natural ecosystems. One more limitation of this study is the isolation of pure bacteria from the E3-degrading consortia on the rich medium LB-agar. This approach does not guarantee isolation of estrogens-mineralizing bacteria due to the absence of selective pressure in the medium, which may explain why none of the isolates could grow on E3 as the sole carbon and energy source.

## Conclusions

We provided solid evidence showing the potential of bacterial consortia to utilize natural estrogens as a carbon source not only in chemically defined medium but also in wastewater and activated sludge microcosms. Estriol metabolism by the IF-consortium proceeds via the 4,5-*seco* pathway, and complete degradation or mineralization of E3 by the IF-consortium probably requires concerted activity of different community members. Although the estrogen degradation efficiency of the IF-consortium was conclusively revealed, it lacks functional robustness, which is one of the main drawbacks of microbial consortia when implemented for environmental applications. To retain the E3 biodegradation phenotype, the IF-consortium must be always handled in the presence of E3.

## ACKNOWLEDGMENTS

This study is a part of a PhD project supported by funds from Arabian Gulf University, Sheikh Zayed bin Sultan Al Nahyan Chair for Environmental Sciences, and University of Bahrain (Bahrain).

## AUTHOR AFFILIATIONS

[1]Allied Health Sciences Department, College of Health and Sport Sciences, University of Bahrain, Manama, Bahrain
[2]Center of Environmental and Biological Studies, Arabian Gulf University, Manama, Bahrain
[3]Biodiversity Research Center, Academia Sinica, Taipei, Taiwan
[4]Faculty of Health and Life Sciences, Northumbria University, Newcastle upon Tyne, United Kingdom
[5]Department of Pharmaceutical Chemistry, Faculty of Pharmacy, Alexandria University, Alexandria, Egypt

## PRESENT ADDRESS

Dorra Hentati, Department of Food, Environmental and Nutritional Sciences (DeFENS), University of Milan, Milano, Italy

## AUTHOR ORCIDs

Wael Ismail ⓘ http://orcid.org/0000-0002-3027-928X
Yin-Ru Chiang ⓘ http://orcid.org/0000-0001-6899-3166
Vartul Sangal ⓘ http://orcid.org/0000-0002-7405-8446

## AUTHOR CONTRIBUTIONS

Jaleela S. Hashem, Data curation, Formal analysis, Investigation, Methodology, Writing – original draft, Writing – review and editing | Wael Ismail, Conceptualization, Funding acquisition, Project administration, Supervision, Validation, Writing – original draft, Writing – review and editing | Yin-Ru Chiang, Conceptualization, Data curation, Formal analysis, Resources, Writing – original draft, Writing – review and editing | Vartul Sangal, Formal analysis, Software, Writing – original draft, Writing – review and editing | Dorra Hentati, Data curation, Formal analysis, Writing – review and editing | Nasser Abotalib, Formal analysis, Methodology | Adnan A. Bekhit, Conceptualization, Writing – original draft, Writing – review and editing

## DATA AVAILABILITY

The trimmed amplicon sequence data have been submitted to SRA under the BioProject accession number PRJNA1214768 and are publicly available. The 16S rRNA gene sequences of the E3-transforming isolates are available in the GenBank under the accession numbers PV018102–PV018106. All other data generated or analyzed during this study are included in this published article and its supplemental information files.

## ADDITIONAL FILES

The following material is available online.

### Supplemental Material

**Supplemental figures and table (Spectrum00741-25-s0001.pdf).** Fig. S1 to S32 and Table S1.

### Open Peer Review

**PEER REVIEW HISTORY (review-history.pdf).** An accounting of the reviewer comments and feedback.

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
