## [Reviewer comments · Microbiology Spectrum]

Microbiology Spectrum

A native bacterial consortium degrades estriol in domestic sewage and activated sludge via the 4,5-seco pathway and requires estriol to retain its biodegradation phenotype

Jaleela Hashem, Wael Ismail, Yin-Ru Chiang, Vartul Sangal, Dorra Hentati, Nasser Abotalib, and Adnan Bekhit

Corresponding Author(s): Wael Ismail, Arabian Gulf University

Review Timeline:

Submission Date:	April 10, 2025
Editorial Decision:	June 6, 2025
Revision Received:	June 30, 2025
Accepted:	July 23, 2025

Editor: Erik Hom

Reviewer(s): The reviewers have opted to remain anonymous.

Transaction Report:

DOI: <https://doi.org/10.1128/spectrum.00741-25>

Re: Spectrum00741-25 (**A native bacterial consortium degrades estriol in domestic sewage and activated sludge via the 4,5-seco pathway and requires estriol to retain its biodegradation phenotype**)

Dear Prof. Wael Ismail:

Thank you for the privilege of reviewing your work. Below you will find my comments, instructions from the Spectrum editorial office, and the reviewer comments.

Please address the reviewer's concerns and revise your manuscript accordingly, making sure to incorporate any explanations that you use to address the reviewers' comments in your revised manuscript. Note that Reviewer #2 has comments provided as an attachment only.

The reviewers note that your manuscript often states conclusions that are not supported by the data presented, and that several conclusions do not state assumptions and are very speculative. Please be conservative in your conclusions: at Microbiology Spectrum, we value sound science - there is no need to exaggerate or oversell your findings. Please turn all unsupported "conclusions" into hypotheses where appropriate. If these problems are not addressed, we will not be able to accept your revised manuscript.

Please also carefully streamline your discussion to present an evidence-based synthesis of your results. This section should be written more clearly and succinctly.

Revision Guidelines

Sincerely,

Reviewer #1 (Comments for the Author):

In this manuscript, the authors focus on identifying several E3-degrading bacterial strains and consortia from raw sewage and activated sludge samples collected from the Tubli Wastewater Treatment Plant (Bahrain). They identify bacteria belonging to the genera *Hydrogenophaga*, *Microbacterium*, and *Gordonia* as E3 degraders. The degradation process is primarily through the 4,5-seco pathway, with potential metabolites identified. Elucidating the molecular mechanisms of E3 biodegradation is important for the development of rational approaches to pollution control. However, I still have significant concerns regarding some aspects of the experimental setup, which lack clarity and scientific soundness. Detailed comments are provided below.

Major comments:

- In the enrichment experiment for E3-degrading bacteria, 1 mM of E3 was added as the selective condition. However, the solubility of E3 in water is 0.119 mg/mL (at 20 {degree sign}C), which is lower than the 1 mM concentration used in the experiment. I have reviewed the response regarding the concern raised in the previous round of review. It appears that the response does not directly address why solubility is not an issue but merely cites other studies that used similar practices, without explaining how this is applicable in the context of the current experiment. Additionally, the references provided do not specifically discuss E3 biodegradation or provide details on the concentration of E3 in aqueous environments. Therefore, the justification for using concentrations above the solubility limit remains unclear.
- Line 263: The statement here seems problematic. While it is true that there is limited literature on the role of raw sewage-borne bacterial consortia in estrogen biodegradation, this rationale does not provide a solid scientific basis for why this consortium is the most appropriate for further study. The microbiomes from raw sewage and activated sludge differ significantly, with the former largely derived from the fecal microbiome. These differences are crucial because microorganisms from raw sewage may not perform well in engineered wastewater systems, where the conditions are different (e.g., varying oxygen levels, nutrient profiles, and the presence of other pollutants). The engineering of these strains for efficient, sustained biodegradation of estriol in activated sludge systems or similar setups remains a significant challenge that should be addressed.
- The manuscript makes several assumptions regarding the temporal dynamics of the IF-consortium in both the E3 and E1 cultures that are not directly supported by experimental data. Specifically, for the E3 culture, data for day 0 and day 2 are missing, with only data from day 4 onward available. Day 0 data is particularly crucial as it would provide insight into the initial community composition, which is essential for understanding how the community evolves over time. The absence of these early-time-point data significantly weakens the conclusions drawn about the consortia's dynamics. Furthermore, for the E1 culture, no data are provided for the first 8 days of the experiment. The authors assume that the compositional dynamics in the E1 culture are similar to those observed in the E3 culture, but this assumption lacks empirical support. Without direct data for the E1 consortia over this period, the claim remains speculative and should be presented as a hypothesis, not as a near-validated conclusion.

Reviewer #3 (Comments for the Author):

The work from Hashem and coworkers focuses on the biodegradation of estriol by bacterial consortia and the importance of such mixed culture in the process. The authors proposed a metabolic pathway for this biodegradation and tentatively identified the bacterial species involved. The revised manuscript accepted some of the suggestions from the previous reviewers, but there are still some issues to be solved.

1. In the abstract section, line 50, according to figure 7B, C. *estronivorus* was dominant starting day 10, both to E1 and E3, and not on day 8.
2. It cannot be understood why the attempt to isolate E3-degrading bacteria was performed without the addition of E3 in the LB agar plates. This would certainly reduce the number of species to test further (line 193)
3. Two different mixtures of estrogens were tested. However, they were called the same, which makes it difficult to differentiate between them in the text (lines 267-268 and further in the text) and in figures, as in Figure 3. Correct.
4. There was a mention of supplementary table S1 that was not provided (line 487).
5. Regarding Figure 7A, the authors replied to one of the reviewers that the composition represents the average of all time points. This is difficult to understand since the cultures supposedly evolved over the incubation time. As such, this figure should have a period that the authors considered relatively stable (defined by a criterion) and then compare the three cultures.
6. *Thermomonas haemolytica* was not specific for E3, just for E1 (line 533)
7. In a general sense, the discussion section is too long and sometimes wordy. There is repetition of text that the authors provided elsewhere, even in the materials and methods section. A more concise and scientific discussion would improve this

paper.

8. The discussion concerning the bacterial population evolution for cultures E3 and E1 was, in part, highly speculative. We cannot forecast what the original composition of the IF consortium was, what happened to culture E1 before day 10, or the same for E3 before day 4. The authors should discuss this part based on experimental data/evidence (lines 705-726).

9. The added paragraph in this new version (lines 787-796) should be deleted, as all of this was already discussed or was present in the conclusions. The same is true for the last paragraph of the conclusions section (lines 820-828).

10. Although the authors reduced the number of figures in the supplementary material section, new ones were also added. Twenty figures of this section were composed of GC-MS spectra (with some with several ones in the same figure). A few ones could be understandable, but not this many. This could be easily replaced by one/two tables that would provide the same data, occupying much less space. This table could even give quantitative values and not just intensity, as the authors were able to quantify the different metabolites, as shown in Figure 5.

In this manuscript, the authors focus on identifying several E3-degrading bacterial strains and consortia from raw sewage and activated sludge samples collected from the Tubli Wastewater Treatment Plant (Bahrain). They identify bacteria belonging to the genera *Hydrogenophaga*, *Microbacterium*, and *Gordonia* as E3 degraders. The degradation process is primarily through the 4,5-seco pathway, with potential metabolites identified. Elucidating the molecular mechanisms of E3 biodegradation is important for the development of rational approaches to pollution control. However, I still have significant concerns regarding some aspects of the experimental setup, which lack clarity and scientific soundness. Detailed comments are provided below.

Major comments:

- In the enrichment experiment for E3-degrading bacteria, 1 mM of E3 was added as the selective condition. However, the solubility of E3 in water is 0.119 mg/mL (at 20 °C), which is lower than the 1 mM concentration used in the experiment. I have reviewed the response regarding the concern raised in the previous round of review. It appears that the response does not directly address why solubility is not an issue but merely cites other studies that used similar practices, without explaining how this is applicable in the context of the current experiment. Additionally, the references provided do not specifically discuss E3 biodegradation or provide details on the concentration of E3 in aqueous environments. Therefore, the justification for using concentrations above the solubility limit remains unclear.
- Line 263: The statement here seems problematic. While it is true that there is limited literature on the role of raw sewage-borne bacterial consortia in estrogen biodegradation, this rationale does not provide a solid scientific basis for why this consortium is the most appropriate for further study. The microbiomes from raw sewage and activated sludge differ significantly, with the former largely derived from the fecal microbiome. These differences are crucial because microorganisms from raw sewage may not perform well in engineered wastewater systems, where the conditions are different (e.g., varying oxygen levels, nutrient profiles, and the presence of other pollutants). The engineering of these strains for efficient, sustained biodegradation of estriol in activated sludge systems or similar setups remains a significant challenge that should be addressed.
- The manuscript makes several assumptions regarding the temporal dynamics of the IF-consortium in both the E3 and E1 cultures that are not directly supported by experimental data. Specifically, for the E3 culture, data for day 0 and day 2 are missing, with only data from day 4 onward available. Day 0 data is particularly crucial as it would provide insight into the initial community composition, which is essential for understanding how the community evolves over time. The absence of these early-time-point data significantly weakens the conclusions drawn about the consortia's dynamics.
Furthermore, for the E1 culture, no data are provided for the first 8 days of the experiment. The authors assume that the compositional dynamics in the E1 culture are similar to those observed in the E3 culture, but this assumption lacks empirical

support. Without direct data for the E1 consortia over this period, the claim remains speculative and should be presented as a hypothesis, not as a near-validated conclusion.

This work reported the enriched E3-degrading bacterial consortia, its degradation properties, the degrading intermediates, the potential degrading pathways, and the change of community structure. This work can enrich the microbial strain sources for the E3 and other estrogens. However, there are several issues need to be explained in its current state.

1. The biodegradation E3 are relatively clear and no new product or pathway is provided. The speculated E3-degrading pathway in this work are not convincing, as the formation sequence of C16 α -OH and C4 α -OH are not sure. Although the results of the time course experiments of degradation are given, only the GC-MS analysis are not enough.
2. Almost of all the degrading intermediates or products were analyzed by GC-MS. It is not very convincing. The metabolites should be isolated, purified and analyzed with high-resolution MS or NMR.
3. Considering the fast transformation efficiency of the consortia for E3 and E2, and very slow for E1, still maintaining very high concentration even after 60 days, it is likely to raise the suspicion that only transformation rather than degradation of estrogenic chemicals happened in the culture of consortia.
4. The growth curves in Fig.1 and 2 are not consistent with each other. Whether there is still other strain functioning in estrogen removal? In Fig.4, the OD reached to 0.8, quite higher than those in Fig. 1 and Fig. 2, and the results in Fig.4A and 4B varied,
5. In Fig. 5, the GC-MS cannot indicated that chemical is C16 α -OH. Why not the hydroxylation at other site? In Fig. 5B, the culture period is very long, much longer than that in Fig.4. There is no negative control. How to prove the reduction of E3 by biolysis not by photolysis during this period?
6. In the diversity analysis of consortia, what is the reason for time points selection? These time points seems no relation with the degrading period. *Croceicoccus estronivorus* dominated the community starting from day 8. No experiment has been performed to study this strain. And it is interesting that it is not consistent with the period of cell growth, E3 reduction and intermediates generation shown in other figures. Please explain it.
7. In Fig.9, The utilization curves of DMSO reflect this consortia may just transform not degrade E3 by using DMSO as energy source.
8. There is no degrading gene or enzyme analysis, which is quite important for the analysis.
9. The data in Fig. S38 are not in accordance with other results. Why?
10. In Discussion, more in-depth discussion regarding the significance and application potentials of this work is warranted with the comparison with other E3-degrading strains and consortia or previous reports.
11. The Conclusion is too long. Shorten and emphasize the importance.
12. There are a lot of grammar and spelling mistakes. The English writing should be improved by a native speaker.

Responses to Reviewers

Reviewer #1 (Comments for the Author):

In this manuscript, the authors focus on identifying several E3-degrading bacterial strains and consortia from raw sewage and activated sludge samples collected from the Tubli Wastewater Treatment Plant (Bahrain). They identify bacteria belonging to the genera *Hydrogenophaga*, *Microbacterium*, and *Gordonia* as E3 degraders. The degradation process is primarily through the 4,5-seco pathway, with potential metabolites identified. Elucidating the molecular mechanisms of E3 biodegradation is important for the development of rational approaches to pollution control. However, I still have significant concerns regarding some aspects of the experimental setup, which lack clarity and scientific soundness. Detailed comments are provided below.

Major comments:

- In the enrichment experiment for E3-degrading bacteria, 1 mM of E3 was added as the selective condition. However, the solubility of E3 in water is 0.119 mg/mL (at 20 {degree sign}C), which is lower than the 1 mM concentration used in the experiment. I have reviewed the response regarding the concern raised in the previous round of review. It appears that the response does not directly address why solubility is not an issue but merely cites other studies that used similar practices, without explaining how this is applicable in the context of the current experiment. Additionally, the references provided do not specifically discuss E3 biodegradation or provide details on the concentration of E3 in aqueous environments. Therefore, the justification for using concentrations above the solubility limit remains unclear.

Response: The publications that we included in our previous responses deal with other estrogens, hence they are within the context of our current study. In addition, there are several values reported for the aqueous solubility of E3 (please see the following sources):

<https://pubchem.ncbi.nlm.nih.gov/compound/Estriol#section=Solubility>

Zhang, C., Li, Y., Wang, C., Niu, L., & Cai, W. (2016). Occurrence of endocrine-disrupting compounds in the aqueous environment and their bacterial degradation: A review. *Critical Reviews in Environmental Science and Technology*, 46(1), 1–59. <https://doi.org/10.1080/10643389.2015.1061881>

If the reviewer wants us to justify the usage of a high E3 concentration, here is the justification: We used environmental samples (sewage, sludge) which naturally harbor a complex diversity of organisms as well as other organic constituents and suspended particles. These may interfere with the availability of E3 to the actual estrogen-degrading bacterial communities. Therefore, addition of concentrations above aqueous solubility limit ensures sufficient supply of E3. Moreover, it is well known that hydrocarbon-degrading bacteria can access the water-insoluble fraction of the hydrophobic substrates via their hydrophobic cell surface. We have now added these explanations to the revised version (Please see lines 165-170).

Please note that the low aqueous solubility of steroids (typically < 1 mg/L) is a well-established physicochemical property that does not precludes bacterial metabolism at millimolar concentrations. While steroid solubility limitations are acknowledged, bacterial degradation of poorly soluble organic compounds occurs through several established mechanisms: (1) direct contact between bacterial cells and solid-phase substrates, (2) localized solubilization at the cell-substrate interface through biosurfactant production, and (3) biofilm-mediated degradation processes. These mechanisms are extensively documented in environmental systems where steroid-degrading bacteria effectively metabolize hydrophobic steroids associated with sediments and soil matrices. The millimolar concentrations employed in this study are consistent with standard protocols in steroid biodegradation research and are necessary to generate sufficient metabolite concentrations for analytical detection and pathway elucidation. Importantly, the primary objectives of this research, namely, enrichment and isolation of bacteria, characterizing metabolic intermediates, and mapping the degradation pathway, rely on qualitative molecular and biochemical analyses rather than quantitative kinetic parameters that would be more sensitive to substrate bioavailability. The growth profiles and degradation metabolite identification remain valid indicators of bacterial degradation capacity regardless of the precise dissolved steroid concentration, as these physiological and molecular signatures reflect the organism's response to steroid exposure and metabolism. The ability of the IF-consortium to degrade mixtures of estrogens in raw sewage and activated sludge at both 1 mM and 0.1 mM concentrations conclusively supports explanations.

- Line 263: The statement here seems problematic. While it is true that there is limited literature on the role of raw sewage-borne bacterial consortia in estrogen biodegradation, this rationale does not provide a solid scientific basis for why this consortium is the most appropriate for further study. The microbiomes from raw sewage and activated sludge differ significantly, with the former largely derived from the fecal microbiome. These differences are crucial because microorganisms from raw sewage may not perform well in engineered wastewater systems, where the conditions are different (e.g., varying oxygen levels, nutrient profiles, and the presence of other pollutants). The engineering of these strains for efficient, sustained biodegradation of estriol in activated sludge systems or similar setups remains a significant challenge that should be addressed.

Response: We did not state that the IF-consortium is the most appropriate for further studies. The rationale for choosing this consortium is the scarcity of knowledge, as stated in the manuscript (Lines 266-267). We agree with the reviewer that the activated sludge and raw sewage microbiomes are different, but this does not necessarily exclude the functionality of raw sewage microbiomes in activated sludge. The fecal microbiome contains steroid hormones-degrading/transforming bacteria, and we showed conclusively that the IF-consortium degraded a mixture of estrogens in activated sludge at two different concentrations.

- The manuscript makes several assumptions regarding the temporal dynamics of the IF-consortium in both the E3 and E1 cultures that are not directly supported by experimental data. Specifically, for the E3 culture, data for day 0 and day 2 are missing, with only data from day 4 onward available. Day 0 data is particularly crucial as it would provide insight into the initial community composition, which is essential for understanding how the community evolves over time. The absence of these early-time-point data significantly weakens the conclusions drawn about the consortia's dynamics. Furthermore, for the E1 culture, no data are provided for the first 8 days of the experiment. The authors assume that the compositional dynamics in the E1 culture are similar to those observed in the E3 culture, but this assumption lacks empirical support. Without direct data for the E1 consortia over this period, the claim remains speculative and should be presented as a hypothesis, not as a near-validated conclusion.

Response: Our assumptions for the temporal compositional shifts in the E1- and E3-degrading communities were based on some facts that we mentioned and explained, and eventually we stated “Nonetheless, these assumptions await validation”. However, as the reviewer still have some concerns, we revised this part in the discussion to address these concerns (Please see lines 513-524; 663-674).

Reviewer #2: No new comments were provided.

Reviewer #3 (Comments for the Author):

The work from Hashem and coworkers focuses on the biodegradation of estriol by bacterial consortia and the importance of such mixed culture in the process. The authors proposed a metabolic pathway for this biodegradation and tentatively identified the bacterial species involved. The revised manuscript accepted some of the suggestions from the previous reviewers, but there are still some issues to be solved.

1. In the abstract section, line 50, according to figure 7B, *C. estronivorus* was dominant starting day 10, both to E1 and E3, and not on day 8.

Response: We stated, "after 8 days". But to avoid confusion, we now corrected it.

2. It cannot be understood why the attempt to isolate E3-degrading bacteria was performed without the addition of E3 in the LB agar plates. This would certainly reduce the number of species to test further (line 193)

Response: Exclusion of E3 from the LB agar plates was proposed to mitigate the selective pressure to facilitate isolation of both E3-transforming and E3-degrading bacteria. However, we fully agree with the reviewer, and we are planning to repeat the isolation even in CDM-agar plates supplemented with E3. We added clarification to the revised version (Please see lines 199-201).

3. Two different mixtures of estrogens were tested. However, they were called the same, which makes it difficult to differentiate between them in the text (lines 267-268 and further in the text) and in figures, as in Figure 3. Correct.

Response: Corrected

4. There was a mention of supplementary table S1 that was not provided (line 487).

Response: Table S1 is an excel file and was already included in the submission. We now added Table S1 to the end of the supplementary material in one file.

5. Regarding Figure 7A, the authors replied to one of the reviewers that the composition represents the average of all time points. This is difficult to understand since the cultures supposedly evolved over the incubation time. As such, this figure should have a period that the authors considered relatively stable (defined by a criterion) and then compare the three cultures.

Response: A comprehensive list of all amplicon sequence variants (ASVs) is generated by the workflow for comparison of species composition between different experiment groups for the Venn diagram. It also includes any singleton ASVs that have only appeared in one sample of the experiment group. We agree with the reviewer that it does not provide insights into variation in microbial composition within an experiment group (different substrates in this study) that is evolving over time. However, the presence of microbial taxa was relatively stable within each substrate group with variation being the abundance of these taxa at different stages (time-points; Fig. 6). The aim of this comparison is to identify taxa that are specific to each substrate. We would apologize for not clarifying this previously and causing confusion.

6. *Thermomonas haemolytica* was not specific for E3, just for E1 (line 533)

Response: The PacBio workflow uses DADA2 algorithm that resolves exact amplicon sequence variants (ASVs) with single-nucleotide resolution from the full-length 16S rRNA gene with near-zero error rate. This led to identification of a few ASVs within some species including *Thermomonas haemolytica*. We previously clarified this in the text “However, DADA2 algorithm defined ASVs based on single-nucleotide difference in the full-length 16S rRNA gene, resulting in multiple ASVs within each species.” We acknowledge the fact that these ASVs are still the same species and have rewritten the information in the manuscript to avoid the confusion (Please

see lines 513-524, 530-538).

7. In a general sense, the discussion section is too long and sometimes wordy. There is repetition of text that the authors provided elsewhere, even in the materials and methods section. A more concise and scientific discussion would improve this paper.

Response: We had to supplement the discussion with more text and explanations to address some comments of reviewer #2. Now, we revised and reduced the discussion length.

8. The discussion concerning the bacterial population evolution for cultures E3 and E1 was, in part, highly speculative. We cannot forecast what the original composition of the IF consortium was, what happened to culture E1 before day 10, or the same for E3 before day 4. The authors should discuss this part based on experimental data/evidence (lines 705-726).

Response: Our assumptions for the temporal compositional shifts in the E1- and E3-degrading communities were based on some facts that we mentioned and explained, and eventually we stated “Nonetheless, these assumptions await validation”. However, as the reviewer still have some concerns, we revised this part in the discussion (Please see lines 663-674).

9. The added paragraph in this new version (lines 787-796) should be deleted, as all of this was already discussed or was present in the conclusions. The same is true for the last paragraph of the conclusions section (lines 820-828).

Response: The added paragraph was requested by one reviewer. The last paragraph in the conclusions was removed.

10. Although the authors reduced the number of figures in the supplementary material section, new ones were also added. Twenty figures of this section were composed of GC-MS spectra (with some with several ones in the same figure). A few ones could be understandable, but not this many. This could be easily replaced by one/two tables that would provide the same data, occupying much less space. This table could even give quantitative values and not just intensity, as the authors were able to quantify the different metabolites, as shown in Figure 5.

Response: We don't have quantification data for the GC-MS analysis. That is why keeping chromatograms is important for clarification. The few figures that were added to the supplementary material were requested by the reviewers.

Re: Spectrum00741-25R1 (**A native bacterial consortium degrades estriol in domestic sewage and activated sludge via the 4,5-seco pathway and requires estriol to retain its biodegradation phenotype**)

Dear Prof. Wael Ismail:

Your manuscript has been accepted, and I am forwarding it to the ASM production staff for publication. Your paper will first be checked to make sure all elements meet the technical requirements. ASM staff will contact you if anything needs to be revised before copyediting and production can begin. Otherwise, you will be notified when your proofs are ready to be viewed.

Sincerely,
Erik Hom
Editor
Microbiology Spectrum